# Respiratory disease in workers handling cross-linked water-soluble acrylic acid polymer

**Takumi Kishimoto**[1]*, **Kenzo Okamoto**[2], **Shigeki Koda**[3], **Mariko Ono**[4], **Yumi Umeda**[5], **Shotaro Yamano**[5], **Tomoki Takeda**[5], **Kammei Rai**[6], **Katsuya Kato**[7], **Yasumitsu Nishimura**[7], **Yoichiro Kobashi**[8], **Tetsuji Kawamura**[9]

**1** Director of Research and Training Center for Asbestos-Related Diseases, Okayama, Japan, **2** Chief of Pathology Hokkaido Chuo Rosai Hospital, Iwamizawa, Japan, **3** Vice Director of National Institute of Occupational Safety and Health, Hadano, Japan, **4** Department of Chemical Inspection, National Institute of Occupational Safety and Health, Hadano, Japan, **5** Department of Pathology, Japan Bioassay Research Center, Kanagawa, Japan, **6** Department of Medicine, Okayama University Hospital, Okayama, Japan, **7** Professor of Radiology, Kawasaki Medical School, Okayama, Japan, **8** Department of Pathology, Tenri Hospital, Tenri, Japan, **9** Vice Director of National Hospital Organization Himeji Medical Center, Hyogo, Japan

* nakisimt@okayamah.johas.go.jp

**Data Availability Statement:** Data cannot be shared publicly because of factories privacy. Data are available from the Japan Organization of

## Abstract

Eight workers involved in packing cross-linked water-soluble acrylic acid polymer, an organic substance, developed pulmonary fibrosis, and the upper lobe was the most affected. The dust concentration in the polymer packing workstation was measured. Chest computed tomography (CT) was obtained for 82 individuals, including the 8 workers mentioned above. Three workers were histopathologically examined. In six of these eight workers, central pulmonary fibrosis and secondary bulla formation caused pneumothorax. Histopathologically, multiple centrilobular fibrotic foci were observed. Chest CT revealed centrilobular nodular opacity and interlobular septal thickening, suggesting early lesions in the workers because the dust concentration was remarkably high. Although the pathogenesis of the disease is unclear, we reported the occurrence of pulmonary fibrosis caused by the exposure to cross-linked water-soluble acrylic acid polymers in humans as it has not been reported earlier.

## Introduction

In April 2017, the Ministry of Health, Labour, and Welfare (MHLW) issued a notice to relevant labor departments to ensure that workers working in polymer industry take health surveys to protect them from exposure to inhaled dust containing cross-linked water-soluble acrylic acid polymer (CWAAP) ("CWAAP" is referred to as "acrylic acid polymer," and all CWAAP, including this acrylic acid polymer, are referred to as "polymers"). This warning was issued after incidences of respiratory disease among workers involved in its manufacturing. Because the pathogenesis of acrylic acid polymer-related lung disease is unknown, the MHLW

Occupational Health and Safety (contact via kenkyu-soudan@honbu.johas.go.jp).

**Funding:** This study was not funded by a commercial company and was financially supported by a grant-in-aid from the Japan Organization of Occupational Health and Safety (Collaborative Research). There is no grant number in this research fund.

**Competing interests:** NO authors have competing interests

requested the Japan Organization of Occupational Health and Safety to conduct a survey and study of respiratory dysfunction caused by acrylic acid polymer dust. Inhaled inorganic dust is a well-known cause of lung tissue fibrosis; however, there are no established results for organic dust, including acrylic acid polymer. Furthermore, no results from animal experiments conducted in the 1990s indicated such extensive and severe pulmonary fibrosis, although focal fibrosis was observed in rats that inhaled polymers for 2 years. However, at least five workers from the same workplace have already been diagnosed with the abovementioned respiratory disease. Moreover, some of them had respiratory disorders from an early age, suggesting that this disease condition is serious. Thus, this study aims to understand the pathology of this disease using data on exposure in the occupational environment by elucidating the clinical and pathological conditions of this disease in present and past workers.

## Methods

Subjects were 83 workers who are handling or had previously handled CWAAP (99% acrylic acid polymer and 1% organic solvent) at a workstation where 5 employees had developed respiratory impairment; they provided their consent to participate in the study. These 83 workers underwent chest contrast-enhanced computed tomography (CT) and completed a questionnaire survey. Of the 83 workers, one refused to undergo a chest CT; therefore, a CT scan was performed on the remaining 82. According to the chest CT results, workers who joined the study in Year 1 underwent annual CT tests over 3 years, whereas workers who joined in Year 2 underwent CT tests over 2 years, and their results were analyzed. Of these, six underwent the first CT scan but did not undergo any subsequent CT tests due to transfer or retirement. Thus, only the first CT test results were analyzed for these six workers. Out of the 82 workers tested, 4 were already being followed up with at a medical institution because their treatment was considered crucial based on the acrylic acid polymer-related findings. Because these four workers provided their consent to participate in the study, chest images taken at their respective medical institutions were interpreted. Workers' tasks involving an acrylic acid polymer are broadly categorized into manufacturing and packing. Packing is broadly categorized into three tasks: filling the manufactured acrylic acid polymer product into drums (Task A); filling the acrylic acid polymer into the hopper to improve the distribution of the acrylic acid polymer filled in the drums (Task B); and the packing task, which comprises placing a fixed quantity of acrylic acid polymer powder into plastic bags in a cardboard box, expelling the air from the plastic bag, closing the bag, and closing the cardboard box with duct tape (Task C) (Fig 1).

Furthermore, the questionnaire survey included questions about how frequently the worker performed. The workers were evaluated based on the frequency and length of time they spent performing such tasks. Workers who performed Task A for ≥5 years and Tasks B and C for ≥6 months were classified as the "high exposure group"; those who performed Task A for <5 years and Tasks B and C for <6 months were classified as the "low exposure group"; and those who had almost no exposure to acrylic acid polymer, such as workers who did not perform Tasks A, B, and C, were classified as the "no exposure group." On May 2018, the questionnaire survey was administered to the respondents, which inquired about the handling of substances, performing specific tasks, years worked, use of protective gear, information related to acrylic acid polymer dust exposure, and smoking history. Testing comprised helical chest CT, including thin slice CT (TSCT). The 5-mm-thick lung field and mediastinal views as well as the 2-mm-thick TSCT lung field views were reconstructed. CT images were observed for the presence of fibrotic changes in the lung field, ground glass opacity, interlobular septal thickening, micronodular opacity, and emphysematous or bullous changes, which suggest peribronchiolar

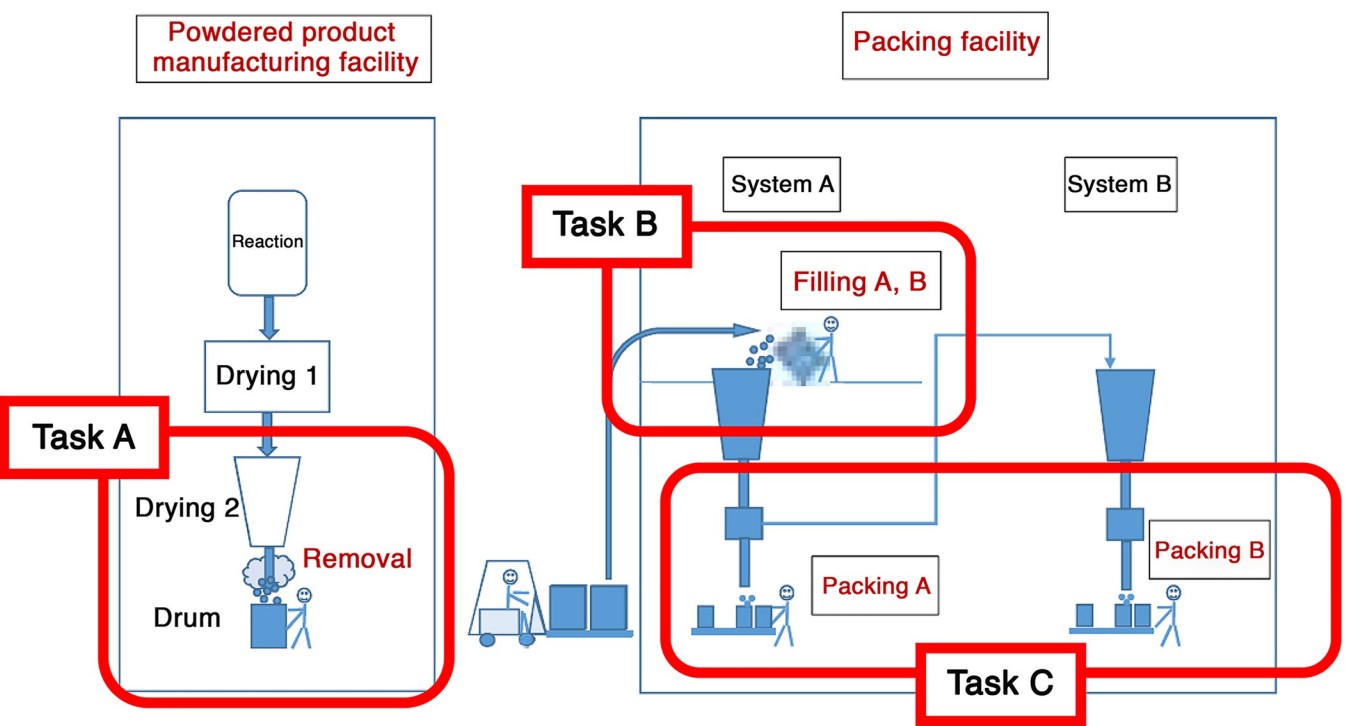

**Fig 1. Schematic representation of the exposure state at the acrylic polymer manufacturing and packing plant.** The polymer powder undergoes a reaction and accumulates in drums after two rounds of drying (production plant). The powder is then packed in containers at the packing plant. Some products are packed in Packing A, while others are packed in Packing B. Workers who were likely exposed to high levels of dust involved in the polymer manufacture and packing. Task B (Filling A, B) involves the highest level of dust, followed by Task C. Task A has a relatively lower exposure than Tasks B and C.

inflammation and mediastinal lymph node swelling. As of Year 1, there was no knowledge of what the initial lesions related to acrylic acid polymer looked like; therefore, even workers with slight findings were selected for follow-up observation, resulting in four categories on images related to the effects of acrylic acid polymer: "findings related to acrylic acid polymer," "suspected findings related to acrylic acid polymer," "slight findings," and "no findings." "Slight findings" were defined as the presence of results that could not be attributed to the inhalation of acrylic acid polymer. These surveys were conducted for 3 years to investigate the degree of progression of the pulmonary lesions over time. Two of the three workers who had already visited a medical institution and undergone surgery or testing underwent surgery for pneumothorax. Another underwent a thoracoscopic biopsy to identify the cause of fibrosis, and their tissue was conserved, so the pathology specimen was obtained with the worker's consent. Histopathologic analysis was performed to compare the findings to pneumoconiosis, such as pulmonary silicosis, or similar diseases, such as sarcoidosis.

## Statistics

For statistical processing, we used a student's $t$-test, and $P < 0.05$ was considered statistically significant.

## Study approval

This study was approved by the 10th research ethics committee of the Japan Organization of Occupational Health and Safety on September 19, 2017 (no. 2). According to the principles of the Declaration of Helsinki, upon fully explaining the purpose of the present study to the

subjects, they were asked to sign a consent form. After response sheets to the questionnaire were submitted, chest CT was performed. We obtained written informed consent.

## Results

In total, 82 workers underwent chest CT scans. All were men who handled acrylic acid polymer in the workplace in the past or present. Workers' ages ranged between 20 and 65 years, with a mean age of 40.5 ± 12.3 years and a median age of 40 years. Fifty were workers in acrylic acid polymer manufacturing, and 32 were primarily involved in acrylic acid polymer packing. Workers in manufacturing worked for 20.2 ± 1.3 days a month for 9.3 ± 7.7 years, and the workers in packing worked for 20.4 ± 2.2 days a month for 3.2 ± 2.0 years. In the past, workers did not wear dust-proof masks for a certain period, but workers currently wear them at all times. Chest CT findings revealed that for the 82 workers with results over the 3 years, there were "no findings (31 workers)," "slight findings (21 workers)," "suspected findings related to acrylic acid polymer (22 workers)," and "findings related to acrylic acid polymer (8 workers)" (Table 1). Of the chest CT results of the 82 workers suspected to be exposed to acrylic acid polymer, central fibrotic findings, ground glass opacity associated with fibrosis, and reticular opacity associated with ground glass opacity were observed in 7, 8, and 8 workers, respectively. Of the 31 workers with intralobular lesions, centrilobular nodular opacity only, interlobular septal thickening only, both centrilobular nodular opacity and interlobular septal thickening, and branching opacity were observed in 12, 9, 9, and 6 workers, respectively. Branching opacity was attributed to smoking rather than acrylic acid polymer in one of the six workers with branching opacity because he had a high smoking index and the distribution was not upper lobe-dominant, distinguishing his results from those of the other 30 workers with an upper lobe-dominant distribution, which will be discussed later. Moreover, pulmonary emphysema was observed in 9 workers, and bullae formation, which can cause pneumothorax, was observed in 19 workers. However, no workers presented mediastinal lymph node swelling, as observed in pulmonary silicosis. Dominance in lesion distribution was observed in 22 workers with "suspected findings related to acrylic acid polymer" and 8 with "findings related to acrylic acid polymer," for a total of 30 workers. All had an upper lobe-dominant distribution (Table 2). Furthermore, of the eight workers with "findings related to acrylic acid polymer," seven had performed Tasks B and C for ≥6 months and one had performed Task A for over 10 years. Moreover, of the 22 workers with "suspected findings related to acrylic acid polymer," 7 had performed Task A and 7 had performed Tasks B and C for ≥6 months, while 6 workers had performed the tasks for <6 months. Two workers had performed acrylic acid polymer manufacturing work at a different work plant. Case 1 worked in the bagging procedure in Task C for 3 years but visited a medical institution for exertional dyspnea and underwent a thoracoscopic biopsy for a detailed examination. Frontal simple chest radiography revealed

**Table 1. Chest computed tomography findings (Synoptical table).**

| CT findings observed throughout the three-year period | | | 82 |
|---|---|---|---|
| | Findings | | 51 |
| | | Slight findings | 21 |
| | | Suspected polymer-related findings | 22 |
| | | Presence of polymer-related findings | 8 |
| | | No findings | 31 |

Based on the CT findings, there are 30 cases of "suspected polymer-related findings" and "presence of polymer-related findings."

**Table 2. Breakdown of computed tomography findings.**

| | | | |
|---|---|---|---|
| Central fibrosis | Yes | N = 7 | (8.5%) |
| | No | N = 75 | (91.5%) |
| Ground glass opacity (fibrosis related) | Yes | N = 8 | (9.8%) |
| | No | N = 74 | (90.2%) |
| Reticular opacity + ground glass opacity | Yes | N = 8 | (9.8%) |
| | No | N = 74 | (90.2%) |
| Intralobular lesions | Yes | N = 31 | (37.8%) |
| Centrilobular nodular opacity | Yes | N = 12 | (38.7%) |
| | No | N = 19 | (61.3%) |
| Interlobular septal thickening (Perilobular only) | Yes | N = 9 | (29.0%) |
| | No | N = 22 | (71.0%) |
| Centrilobular nodular opacity+ interlobular septal thickening (centrilobular + perilobular) | Yes | N = 9 | (29.0%) |
| | No | N = 22 | (71.0%) |
| Branching opacity | Yes | N = 6 | (19.4%) |
| | No | N = 25 | (80.6%) |
| No findings | No | N = 51 | (62.2%) |
| Pulmonary emphysema | Yes | N = 9 | (11.0%) |
| | No | N = 73 | (89.0%) |
| Bullae | Yes | N = 19 | (23.2%) |
| | No | N = 63 | (76.8%) |
| Mediastinal lymph node swelling | Yes | N = 0 | (0.0%) |
| | No | N = 82 | (100.0%) |
| Dominance of lesion distribution | Yes | N = 30 | (36.6%) |
| Upper | | N = 30 | (100%) |
| Middle | | N = 0 | (0%) |
| Lower | | N = 0 | (0%) |
| Diffuse | | N = 0 | (0%) |
| No findings | No | N = 52 | (63.4%) |

Eight items of CT findings for 82 cases are described in this table.

bilateral decreased pneumatization in the upper lung fields and an upward shift of the pulmonary hila (Fig 2). Moreover, chest CT results showed severe central fibrosis, upper lobe nodular opacity, ground glass opacity with bullae formation, and interlobular septal thickening (Fig 3). These typical results were observed in four other cases, including case 2, whose chest radiography is shown in Fig 4. The five cases with typical imaging results worked in acrylic acid polymer packing from 26 months to 3 years (median: 2.75 years), which is an extremely short time period. Case 3 worked on the filling task (Task B) for 2 years but had no subjective symptoms. Even in patients with only mild fibroses on chest CT, central fibrosis, shadows (mainly ground glass opacity), and bullae formation were observed (Fig 5). Case 4 worked in acrylic acid polymer manufacturing for 20 years and had a history of surgery for a right pneumothorax. On chest CT, this case, who had even milder fibrosis, had bullae formation as well as mild ground glass opacity and interlobular septal thickening (Fig 6), but as mentioned later, characteristic histopathological findings of this disease were observed. We further speculated that such cases with micronodular opacity and interlobular septal thickening on chest CT (Fig 7) could be early lesions and were followed up on as suspected cases. Then, we investigated the relationships between the degree of exposure and chest CT results in the acrylic acid polymer manufacturing and packing tasks (Fig 8). In terms of packing, chest CT results with suspected

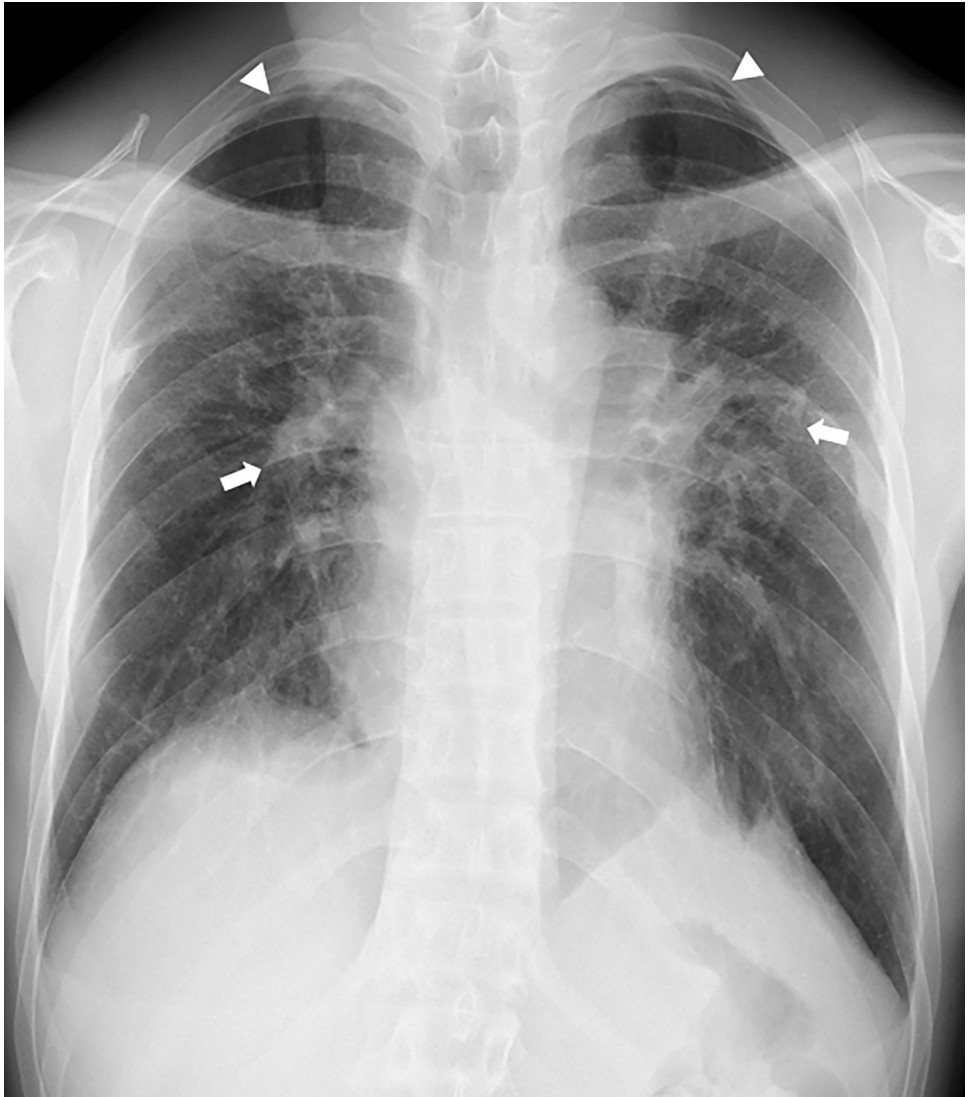

**Fig 2. Typical frontal chest X-ray image of Case 1.** Chest radiography (Case 1). The bilateral upper lung field decreased, and fibrosis was observed in the hilar (arrow) region. Multiple bullas are in the bilateral apex (round arrow).

and certain links to acrylic acid polymer were observed at a higher rate in workers who were involved in the filling task and thus exposed to higher levels of the chemical. We believe there is a correlation between the degree of exposure and the disease state or early lesions in this disease. Cases 1–3 presented typical imaging results of the disease. Histopathological specimens obtained by thoracoscopic biopsy in these cases showed numerous centrilobular fibrotic foci of the respiratory bronchioles (alveolar duct space), perialveolar intraluminal obliteration, and wall fibrosis (Figs 9–12). These foci showed intense fibrotic changes and high levels of intense fibroblastic hyperplasia but only slight or no inflammatory cell infiltration. On polarized microscopy, no dust deposits or granulomas were observed. Overexpansion was observed in the peripheral airspace. Moreover, thickening and fibrotic changes of the axial connective tissue pleura and interlobular septa were observed. There were almost no background alveolar wall changes; only lymphocyte aggregation and lymphoid follicles were observed in certain regions. These centrilobular lesions suggest a high likelihood that the lesions are related to

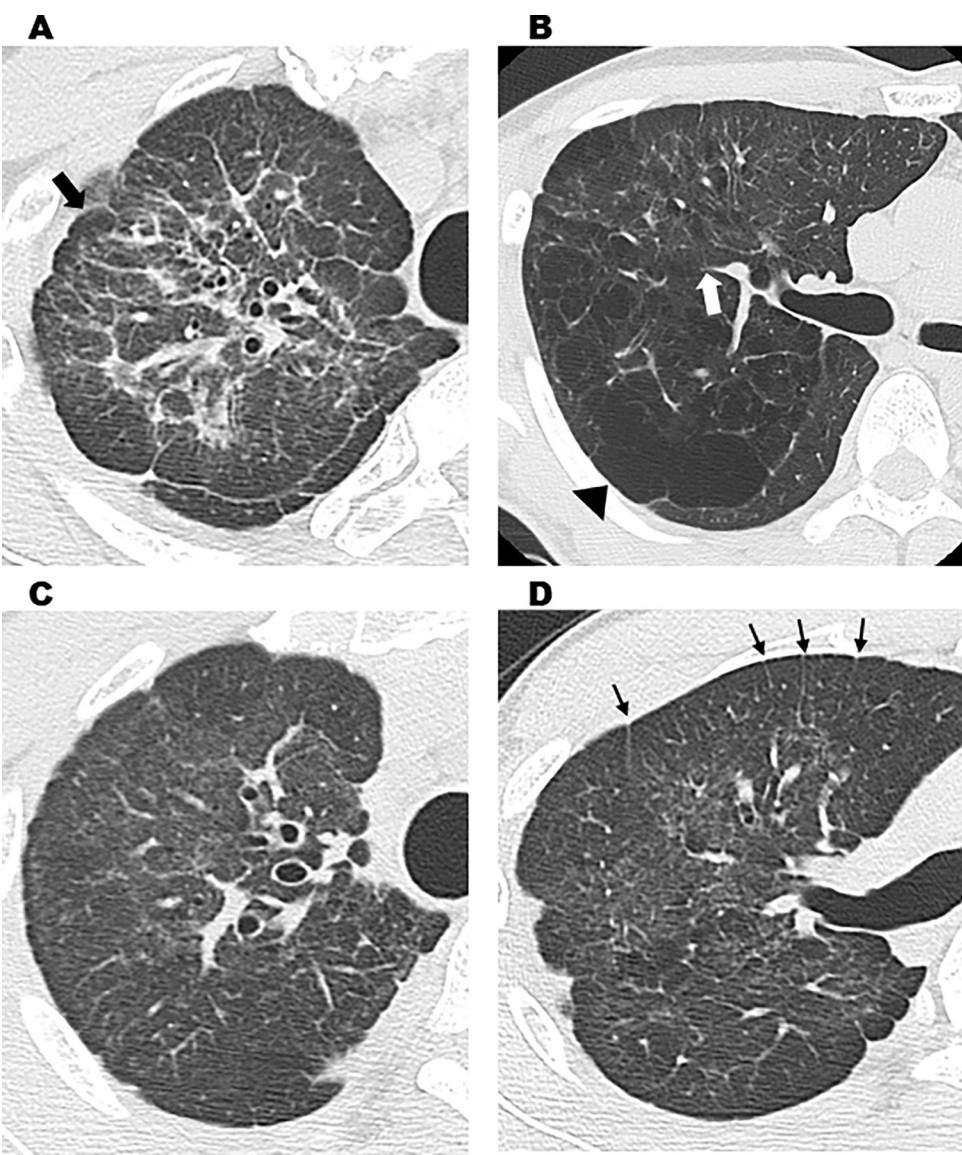

**Fig 3. Chest HRCT of Case 1.** (A) HRCT image (Case 1). Severe central fibrosis was observed on the right upper lobe. Fibrosis involved the pleura, and subpleural bullae formation (large arrow) was observed. (B) Emphysematous changes (round arrow) thought to be related to bronchial obliteration are observed. The white arrow indicates the obstruction of the bronchus. (C) In areas of mild fibrosis, ground glass opacity can be observed. (D) Emphysematous changes and interlobular septal thickening (small arrow) are observed.

inhalation, possibly pneumoconiosis caused by a specific inhaled substance. Although the specimens were collected from the peripheral tissues of the lung, tissue from the central parts could not be collected; therefore, the central changes were unknown. The histopathological results of specimens collected from the surgery for pneumothorax in case 4 showed subpleural emphysematous bullae formation (Figs 13 and 14). Although not abundant, the abovementioned centrilobular fibrotic foci were observed along with fibrotic thickening of the pleura of the axial connective tissue and interlobular septa. The CT imaging results revealed severe central-dominant fibrosis of the peribronchial tissues of the upper lobe within merely 3 years of inhaling acrylic acid polymer dust; however, the histopathological results of the central parts were not included in the collected specimen, so their details were unclear.

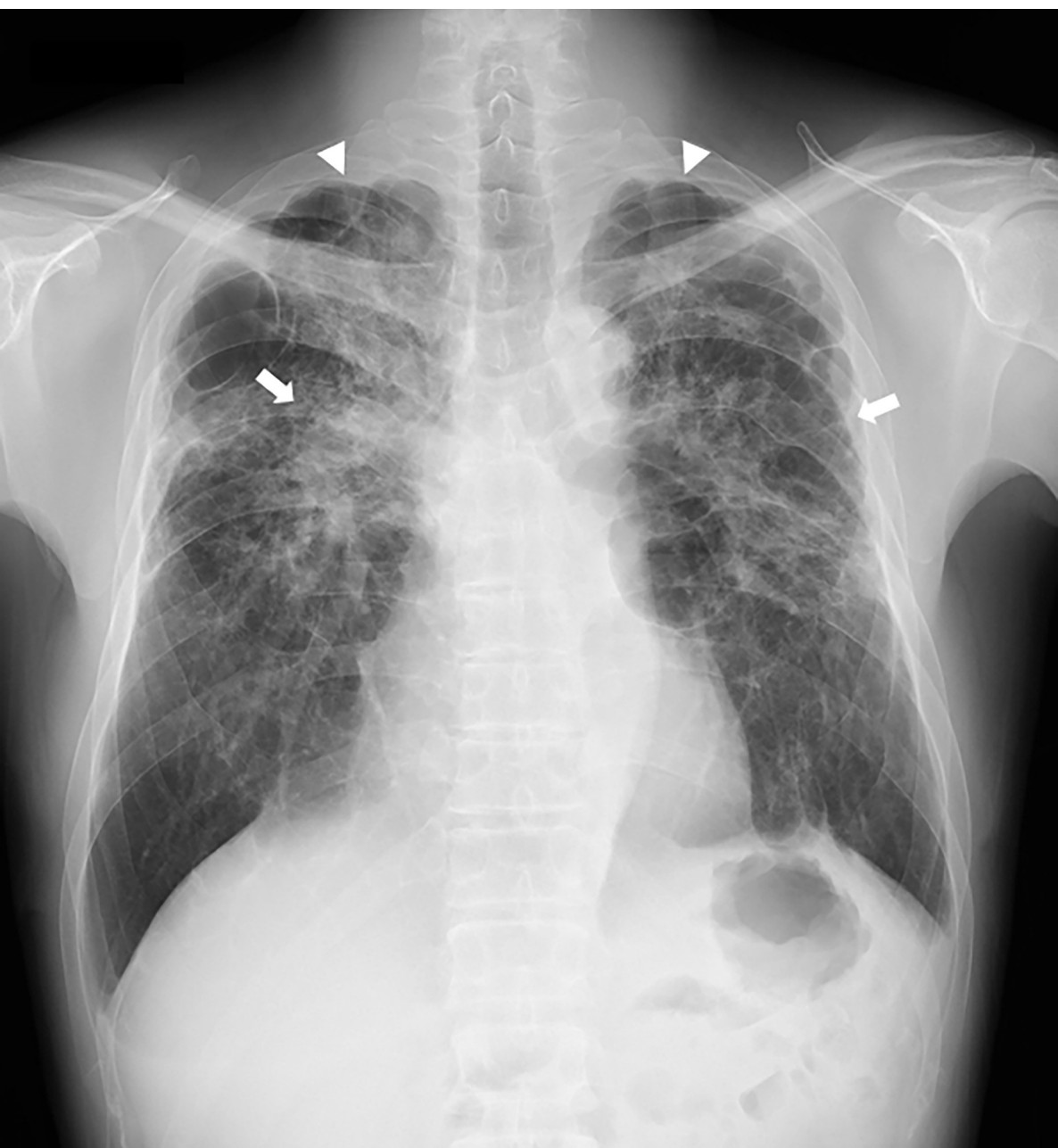

**Fig 4. Typical frontal chest X-ray image of Case 2, chest radiography.** The bilateral upper lung field (round arrow) decreased due to elevation of the lower lobes caused by bilateral upper lobe fibrosis and pulmonary apical bulla formation, and fibrosis in central hila and fibrosis (arrow) as well as bulla formation (round arrow) can be observed. The bilateral inferior lobes are lifted up to the upper portion because of the reduction in the upper lung.

## Discussion

Acrylic acid polymer, the subject of this study, is a high-molecular-weight compound of the monomer acrylic acid and has a cross-linked structure, which is obtained by reacting the polymer with a cross-linking agent. It is used as a chemical intermediate in the production of pharmaceuticals and cosmetics (acrylic acid polymer contained in the final products, such as pharmaceutical products and cosmetics, does not return to the original inhalable dust form).

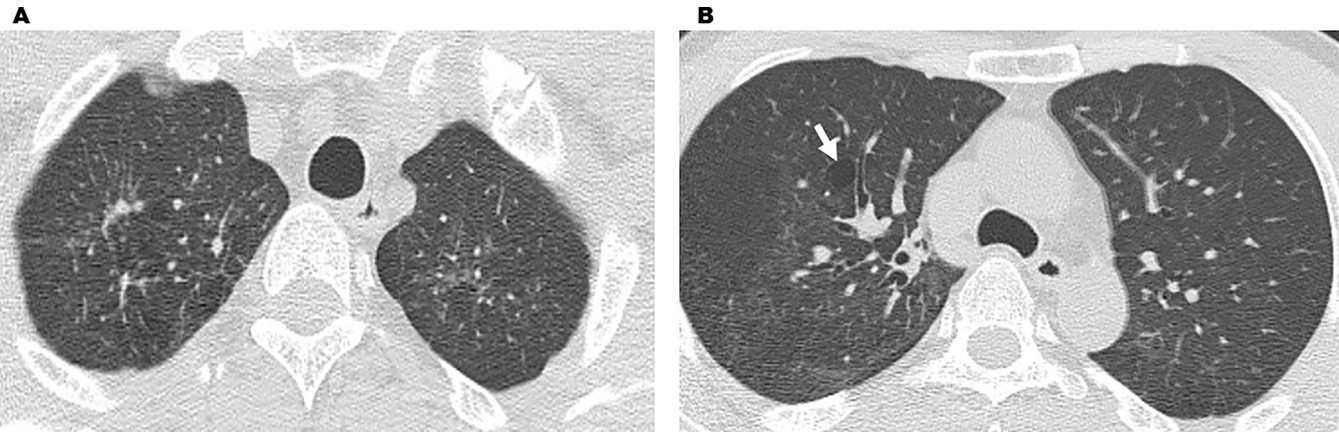

**Fig 5. HRCT image of Case 3 with a definite lesion caused by polymers.** (A) Image of fibrosis of the bilateral apices. The severity of fibrosis is moderate, with ground-glass opacity. An area of the upper lobe showed focal emphysema caused by bronchial constriction. Some areas with peripheral bullae and emphysematous changes were also observed. (B) Image of localized emphysema of the right upper lobe. Central peripheral bullae formation (arrow) and emphysematous changes can be observed (Case 3).

The diameter and shape of acrylic acid polymer microparticles inhaled by humans or animals change as they pass through the airway and come into contact with moisture. Moreover, it is thought to transition from aqueous dispersed or swellable particles to thickened gels, depending on the fluid composition, pH, and other environmental factors. They are clearly distinct from inorganic dusts, such as crystallized silica and asbestos, which exist as solids in the lungs. Therefore, the toxic form of acrylic acid polymer in the bronchial, bronchiolar, and alveolar microenvironment rich in water, phospholipids, and mucus is unknown, and whether it can even be identified in lung tissues is unknown. Furthermore, as described in this study, the clinical and histopathological characteristics of the acrylic acid polymer lung are extremely different from those observed in diseases caused by organic dusts, such as bronchial asthma or alveolar hypersensitivity. This explains why it is important to compare it to the previous studies of pneumoconiosis. There is very little information or research on the adverse events caused by polymers. However, since the late 1980s, epidemiological studies on workers involved in polymer manufacturing have been conducted [1], but there have been no reports on health damage or respiratory disorders related to polymer exposure [2]. Furthermore, in animal studies, rats systemically exposed to inhaled polymer for 2 years at 0.8 mg/m$^3$, a low

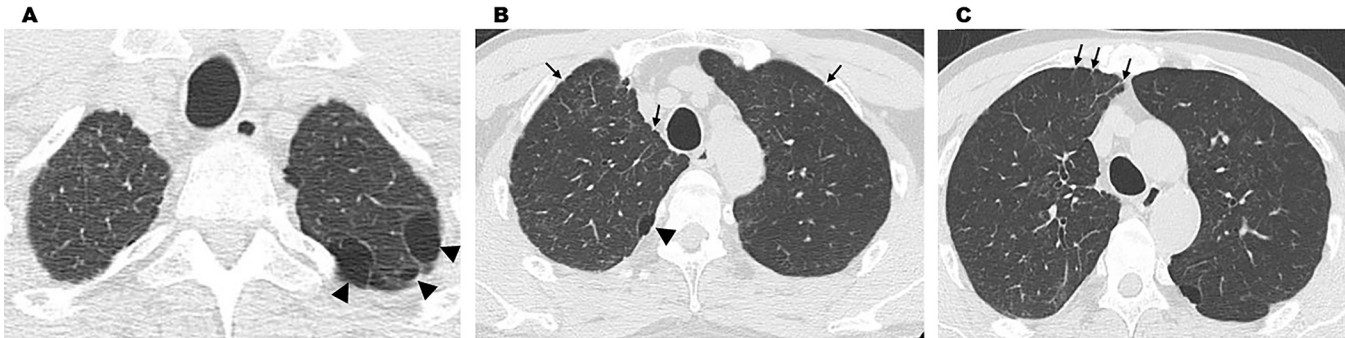

**Fig 6. HRCT image of an early-stage case with a history of pneumothorax.** (A) Multiple bullae (arrow) of the bilateral apices. (B) Image of interlobular septal thickening (arrow). (C) Interlobular septal thickening and bulla (round arrow) directly beneath the pleura. Case with a history of pneumothorax. Bilateral centrilobular nodular opacity at the apices and numerous bulla formation are observed. Interlobular septal thickening is observed (arrow). This case had only polymer manufacturing experience and no packing experience (Case 4).

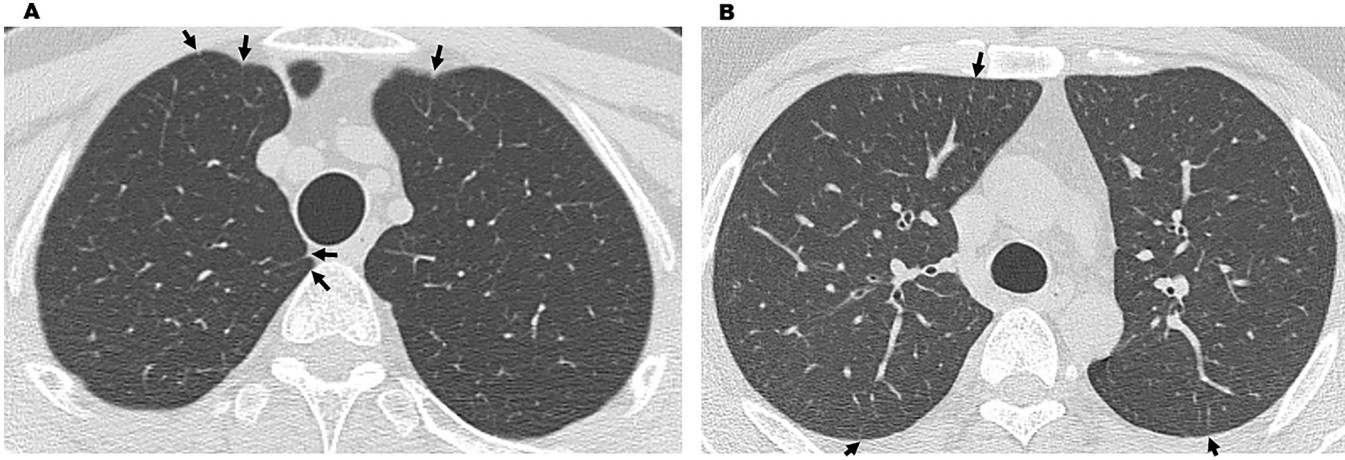

**Fig 7. HRCT image of an early-stage case.** (A) Interlobular septal thickening. (B) Interlobular septal thickening. Chest CT image revealing suspected polymer lesions. Bronchiolar and bronchial inflammations and interlobular septal thickening (arrow) are observed.

level of exposure, developed lung fibrosis. Another study of systemic inhaled exposure in rats for up to 52 weeks found alveolar inflammation that worsened with the concentration and duration of exposure. However, inflammation decreased in a group that was subjected to a recovery period after exposure, and some rats recovered completely depending on the concentration [3–5]. When an application for workers' accident compensation for this disease was received, the MHLW made a request to the National Institute of Occupational Safety and Health of Japan (JNIOSH), which visited the workplace and performed an occupational hazard investigation to investigate the cause of the disease. In this case, the NWPS-254 sampler used for collection conforms to the ISO 7708 standards and allowed for selective sampling of respirable particles, defined as particles that can reach pulmonary alveoli; therefore, we measured Tasks B and C for acrylic acid polymer manufacturing work and acrylic acid polymer packing work. Individual inhalable dust exposure in Task B ranged from a minimum concentration of

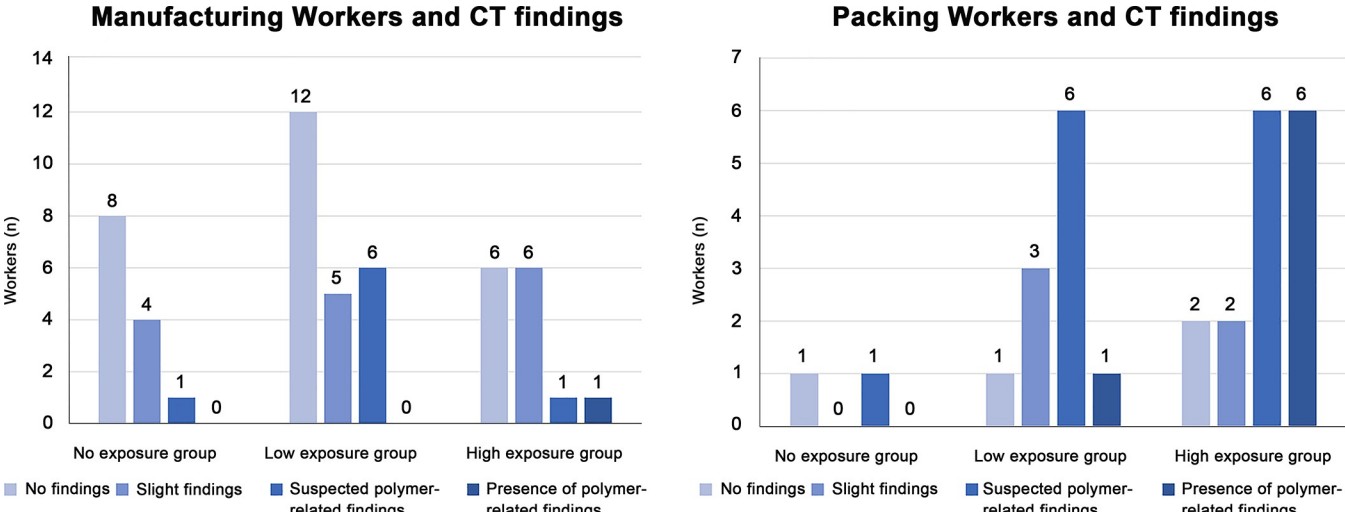

**Fig 8. Incidence of polymer findings on chest imaging according to different exposure intensities.** Proportions of exposure intensity and findings. (The CT results for packing workers (A, B, C) do not include the three workers who did not answer the question about exposure severity.).

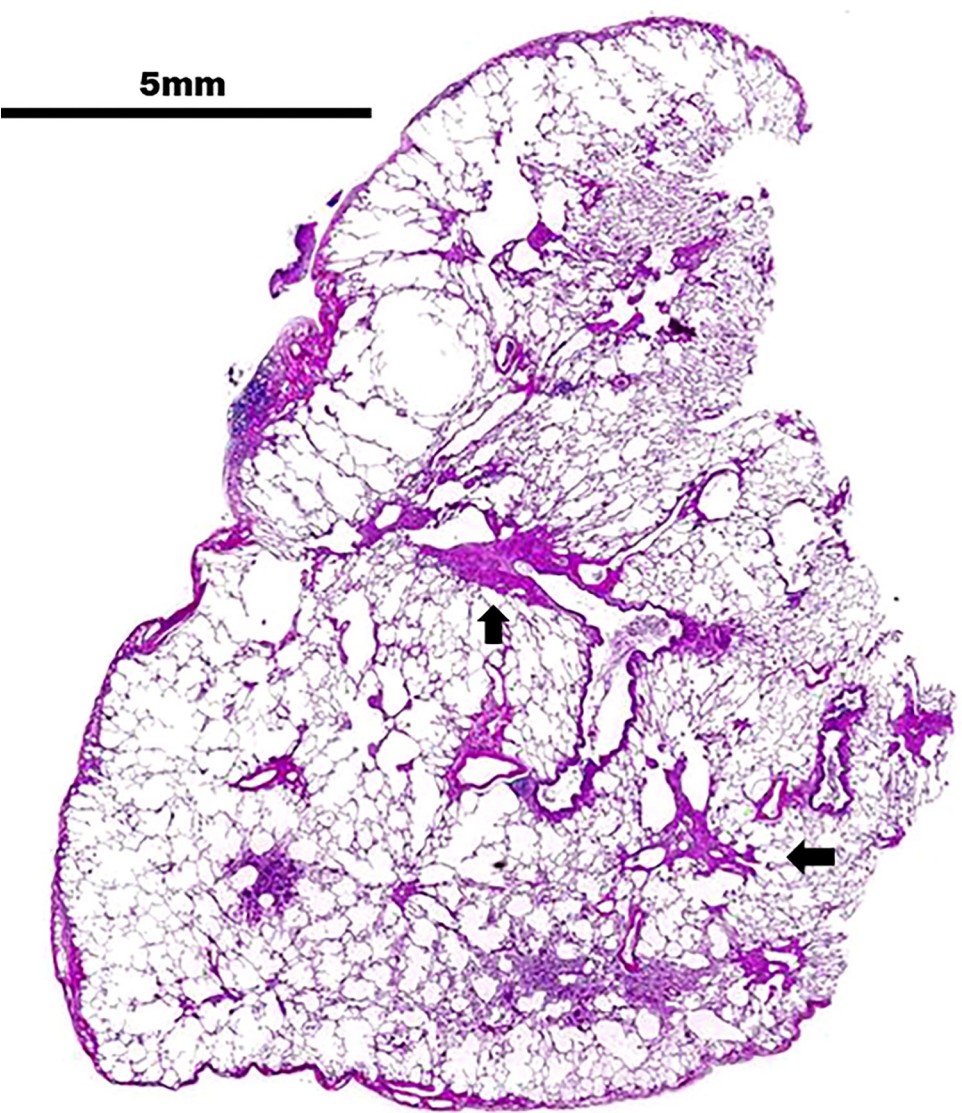

**Fig 9. Magnified video-assisted thoracoscopic surgery view of the histopathological findings of Case 1 obtained by biopsy.** Case 1. Specimen magnified video-assisted thoracoscopic surgery view at left S1 + 2. Irregular foci interspersed (arrow) with perifocal airspace dilation (overexpansion) are observed.

4.2 mg/m$^3$ (mean concentration throughout the 31-min task) to a maximum concentration of 41.8 mg/m$^3$ (mean concentration throughout the 22-min task). The location where Task B was performed was enclosed to prevent dust from dispersing, which may have increased the dust concentration because of a smaller working space volume. Task C had a minimum concentration of 4.0 mg/m$^3$ (mean concentration over 5 h 31 min) and a maximum concentration of 5.9 mg/m$^3$ (mean concentration of 1 h 14 min). Individual 8 h-time weighted average (TWA) of inhalable dust of workers in Task C ranged between 0.4 and 1.4 mg/m$^3$, resulting in an 8 h-TWA of 3.2–7.6 mg/m$^3$ for workers involved in both Tasks B and C. Moreover, respirable dust, which can be inhaled into the deep tissues of the lungs, accounted for 20%–30% of all inhalable dust [6]. In this study, for 3 years, we screened acrylic acid polymer packing workers who received compensation for workers' accidents by chest CT. The interpretation of our results was similar to that of the occupational health survey that was conducted by the

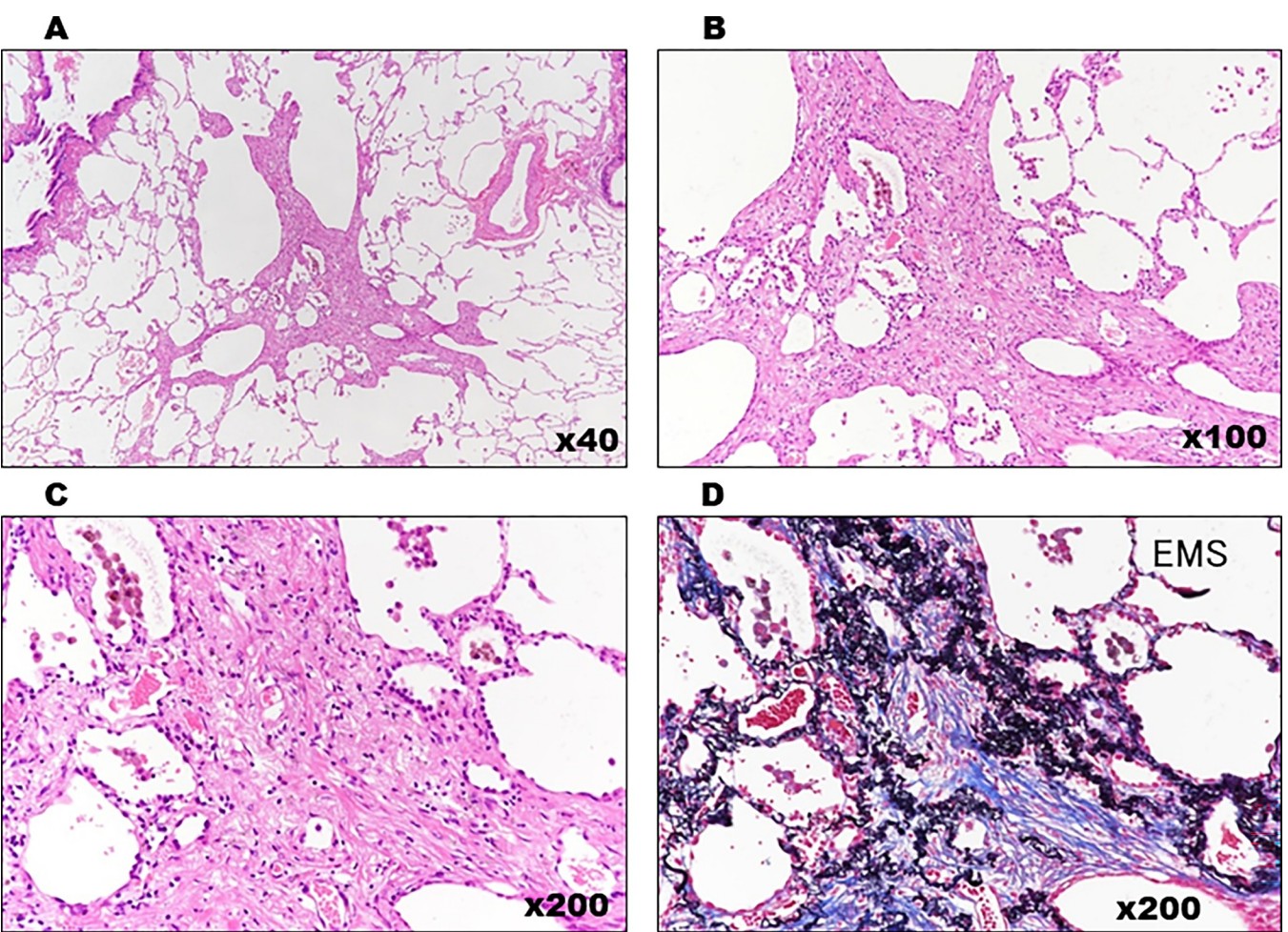

**Fig 10. Histopathological view of Case 1.** (A–D) Medium-to-intense magnification of central lobulette fibrotic foci with reduced luminal obliteration and wall fibrosis. The basic structures of the lungs are preserved. Fibroblast hyperplasia with mild inflammatory cells and the absence of dust deposits are observed. (EMS: Elastica Masson Stain).

JNIOSH. A general interpretation of the chest CT and histopathological results of the 82 workers revealed that cases with the primary features of central fibrosis, micronodular opacity with bullae formation and peribronchiolar fibrosis, and interlobular septal thickening were typical results of respiratory disease related to acrylic acid polymer. Central fibrosis was bilateral but more intense in the right upper lung field. Upper lobes showed bilaterally decreased pneumatization and an upward shift of the hila. This result is similar to pulmonary silicosis with large opacities. Evidently, it differed from progressive massive fibrosis in that it did not comprise fibrotic lesions that formed lumps but rather intense fibrotic lesions with central dominance that decreased lung volume. In the peripheral pathological tissues, centrilobular fibrotic foci as well as pleural and interlobular septal thickening were observed. The centrilobular fibrotic foci did not resemble histiocyte/fibroblast hyperplastic fibrosis associated with dust deposits primarily on the respiratory bronchioles and alveolar duct walls, nor did they resemble the mixed dust fibrosis of normal pneumoconiosis or dust macules [7]. Granuloma was not observed. Tissues from central lesions could not be obtained from the cases that were analyzed in this study, and acrylic acid polymer, which could be the cause of peripheral lesions, was not detected. The relationships between these lesions and the disappearing and undetectable

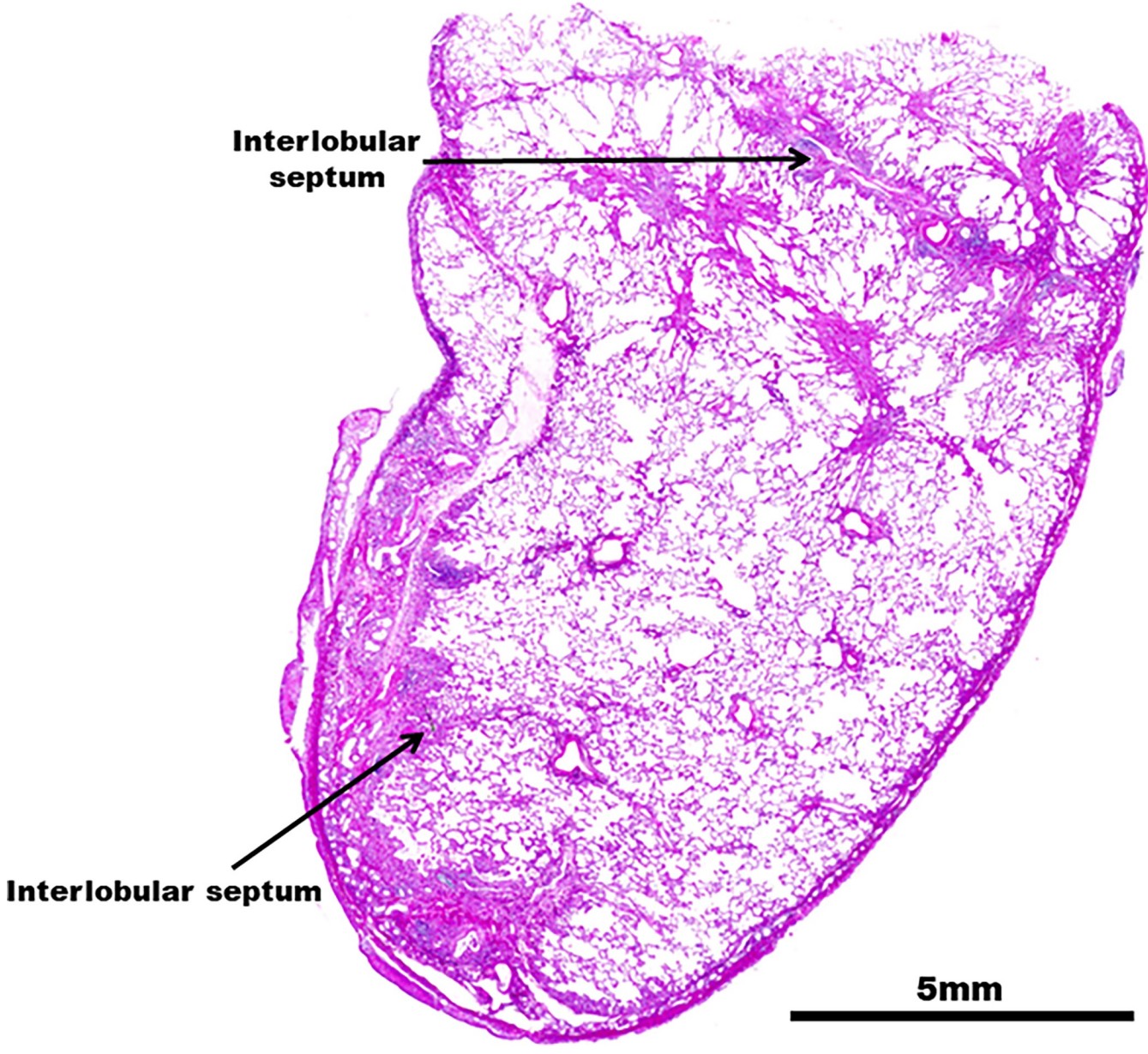

**Fig 11. Magnified video-assisted thoracoscopic surgery view of the histopathological findings of Case 2 obtained by biopsy.** Case 2. A magnified video-assisted thoracoscopic surgery view of the left S3 shows interspersed irregular foci, perifocal airspace dilation (overexpansion), and pleural and interlobular septal thickening (arrow).

acrylic acid polymer will be explored in the future. Our cooperating research agency, the Japan Bioassay Research Center, has been following up on this topic in an animal exposure study of inhaled acrylic acid polymer, which is expected to add to our understanding of this relationship. In this study, while relatively weak fibrosis on chest CT, we focused on cases with centrilobular micronodular opacity in terms of fibrosis or interlobular septal thickening. Centrilobular micronodular opacity was observed in 19 cases, 9 of which also had centrilobular fibrotic foci and interlobular septal thickening. However, Cases 2 and 3 had early fibrosis,

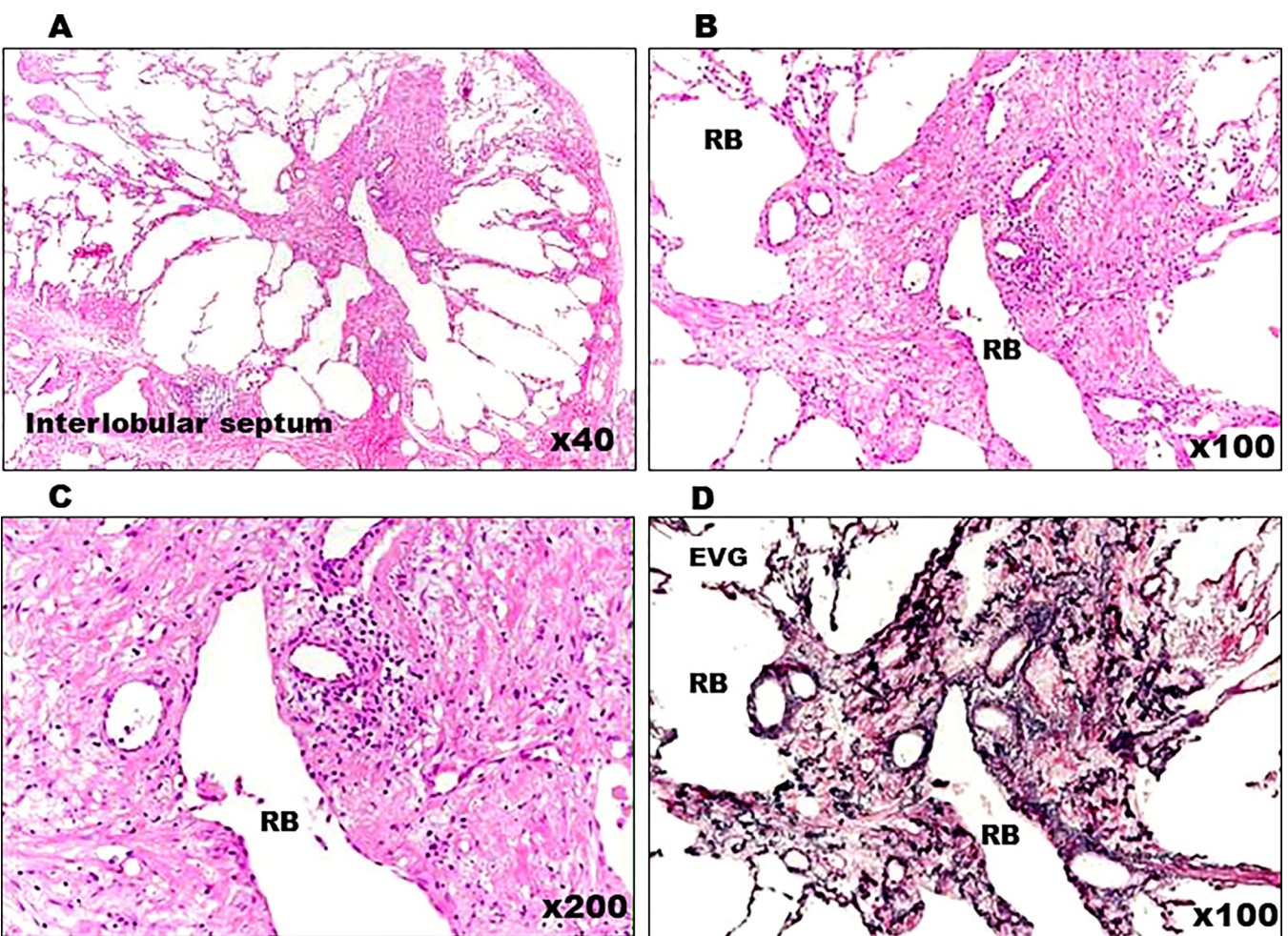

**Fig 12. Histopathological view of Case 2.** (A) Medium-to-intense magnification of central lobulette fibrotic foci with reduced luminal obliteration and wall fibrosis. (B–D) Mild inflammatory cell infiltration with the overexpansion of peripheral airspace (B, C) and the absence of dust deposits (D). (RB: respiratory bronchioles, EVG: Elastica van Gieson).

indicating a high likelihood that this interlobular septal thickening is related to acrylic acid polymer and providing important evidence for lung damage caused by an acrylic acid polymer. Moreover, branching opacity, which represents bronchiolar inflammation, was observed in six cases. These three results suggest that an acrylic acid polymer that passes through the bronchioles and the alveolar duct into the alveoli may be processed by the alveolar macrophages and then by the lymphatic tracts of the axial connective tissue. As in Cases 1 and 2, images and pathological results from several cases show that the respiratory bronchioles are obliterated and constricted by fibrosis and that their peripheral tissues are overexpanded or emphysematous. Therefore, bullae formation is prone to occur in this disease, forming emphysematous bullae in certain cases. Moreover, the progression of such fibrosis results in bilateral central-dominant fibrosis of the upper lobes, decreasing lung capacity, and causes clear pulling of the pleura and lung, which results in bullae formation in the walls and pneumothorax. Bullae formation was observed in 16 cases of this disease (19.5%), 4 of which were complicated by pneumothorax. Workers with bullae formation should be warned about the risks of developing a pneumothorax. The fibrosis results shown above are bilateral and dominant in the upper lobes; i.e., the lower lobes were spared, preserving respiratory function, and none of the cases

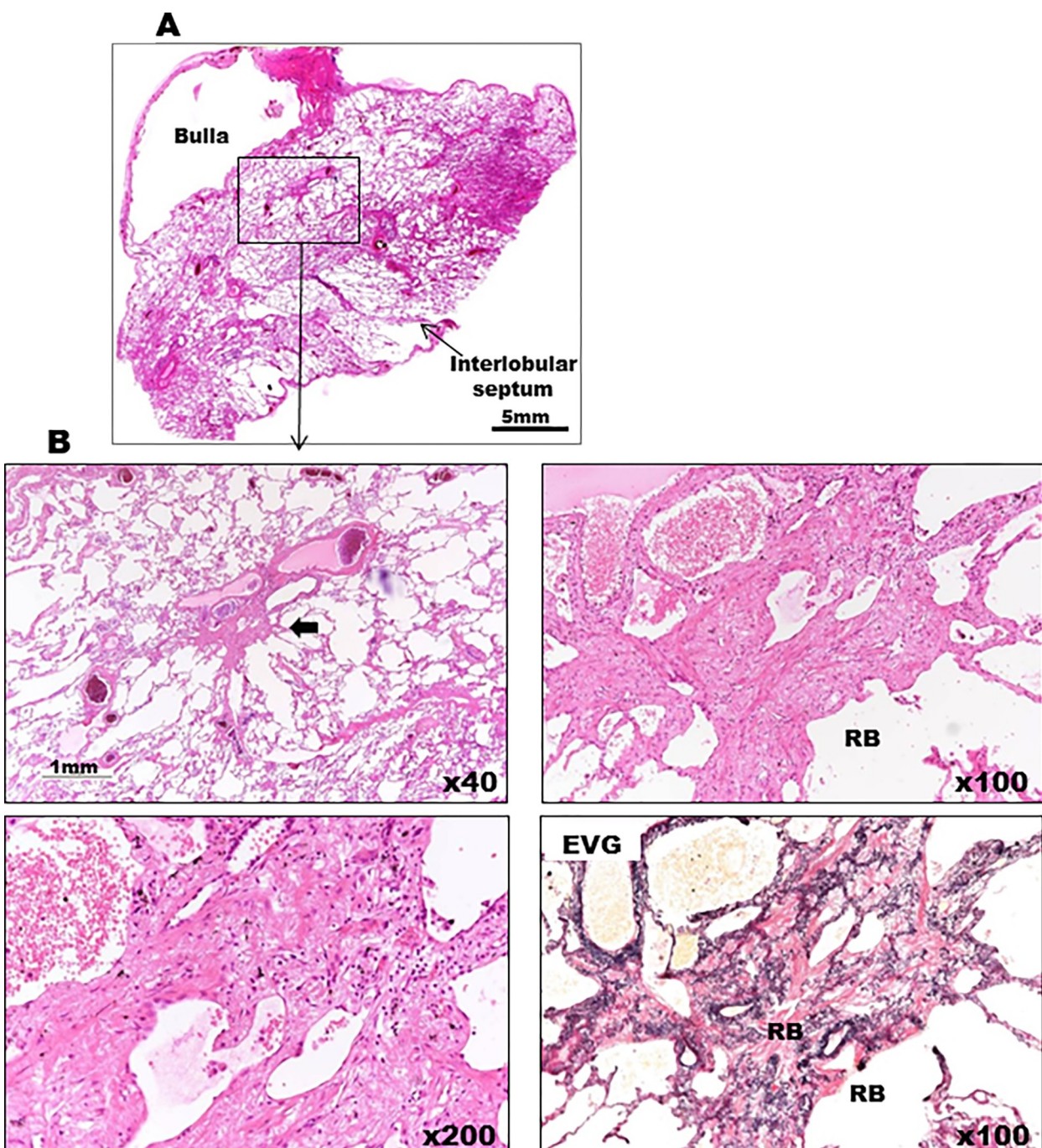

**Fig 13. Magnified view of the histopathological findings of Case 4 obtained by pneumothorax surgery (top picture).** (A) Emphysematous bullae formation and interlobular septal thickening are observed. (B–E) Histopathological findings of Case 4 are shown. Medium-to-intense magnification of central lobulette fibrotic foci, reduced luminal obliteration, and wall fibrosis with insignificant inflammatory cell infiltration.

had significant respiratory dysfunction, which is one of the conditions for receiving compensation for pneumoconiosis caused by an occupational hazard. The distribution of these lesions as well as the right-dominant laterality and centrilobular location of the fibrotic foci were similar to pneumoconiosis of dust particles, as seen in pulmonary silicosis, suggesting that this disease

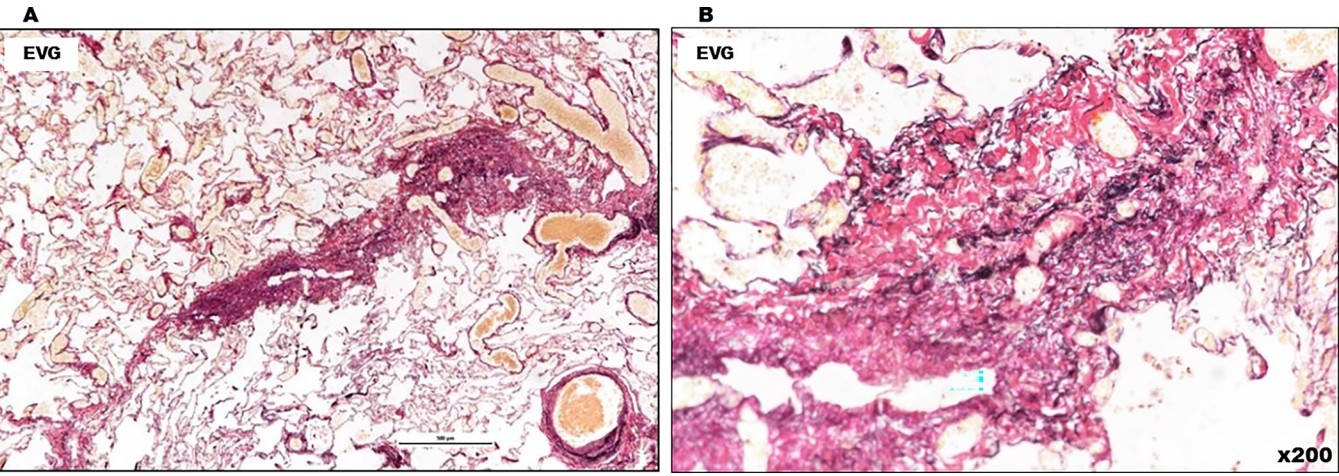

**Fig 14. Histopathological findings of Case 4.** (A, B) Fibrotic thickening of interlobular septa is observed.

was caused by inhalation of the microscopic and light acrylic acid polymer dust. Additional investigation is needed to determine whether such early lesions cause central fibrosis in a relatively short time period. Among the imaging results of workers in packing tasks involving acrylic acid polymer (Tasks B and C), fibrotic lesions were observed in 7 out of 8 workers (87.5%), and 14 out of 22 workers (66.7%) were suspected of having fibrotic lesions. In addition to these results, it is important to consider the results of the abovementioned JNIOSH survey and the fact that the five cases in packing who were approved to receive compensation for occupational health damage only worked in packing for a short time period, ranging from 26 months to 3 years [8]. It is assumed that short-term and consistent exposure at these concentrations caused the lung disease. As previously reported, another type of CWAAP inhaled by F344 rats at a concentration of 10 mg/m$^3$ for 13 weeks can generate neo-connective tissues, suggesting fibrosis; this topic requires further investigation in an animal test that we are planning to conduct. Moreover, among the workers involved in the manufacturing of acrylic acid polymer, one was found to have fibrotic results (2%) and eight were found to have suspected fibrotic results (16%), both of which were lower than the rates observed among those involved in packing tasks. The exposure time was up to 15 min for the manufacturing task, and it is estimated that they were not involved in any other task that exposed them to acrylic acid polymer. Thus, it can be considered a low-dose exposure. The one case with fibrotic results involved in acrylic acid polymer manufacturing was exposed to low-dose but long-term exposure because he worked on the task for 20 years. Moreover, eight cases were suspected with fibrotic results, suggesting that prolonged low-dose exposure can cause lung disease. As mentioned above, almost all cases of fibrosis or interlobular septal thickening were exposed to high levels of acrylic acid polymer for short durations of 2–3 years or were exposed at low doses for 20 consecutive years, which suggests a dose-dependent relationship in this disease. However, normal pulmonary silicosis does not cause peripheral emphysematous changes related to central- or bronchiole-dominant fibrosis-associated airspace obliteration. For sarcoidosis, fibrosis progresses through the lymphatic tracts [9]; however, bronchiolitis is not generally observed and differs from this disease. After being processed in the alveoli, acrylic acid polymer passes through and is processed in the lymphatic tracts. This is similar to pulmonary silicosis, but unlike pulmonary silicosis or sarcoidosis, there were no cases of mediastinal lymph node swelling, which appears to distinguish the diseases. It is unclear whether working with dust for 2–3 years reduces vital capacity due to intense, central-dominant fibrosis. The dust we examined

in this study was organic and was previously not reported to cause pneumoconiosis-related lesions. However, to prevent such respiratory diseases, it is important to improve the occupational environment and to wear appropriate dust-proof masks with protection factors of 50 or higher (e.g., powered air-purifying respirators) because there are no designated occupational exposure limits.

For acrylic acid polymer packing and manufacturing work, we reported that central fibrosis is characterized by the onset of pulmonary lesions. After a chest CT examination, we confirmed typical pulmonary fibrosis in eight workers. Furthermore, in 22 out of 82 workers, we observed centrilobular nodular opacity or ground glass opacity and interlobular septal thickening, which were thought to be early-stage lesions. In the packing work of such plants, the concentration of polymer dust was extremely high. Moreover, we reported a dose-response relationship, in which the longer the hours spent working in a workplace with a high degree of exposure, the higher the frequency of lesions. However, the mechanism underlying the onset of this pathology is unclear, and we believe that additional elucidation will be required through animal experiments.

## Acknowledgments

This workplace has made safety and sanitation improvements to the work place and procedures, such as modifying the packing areas, before this study was started. They have introduced the use of electric air-purifying respiratory protective gear. We would like to express our gratitude to the plant, which provided generous cooperation throughout this study period.

## Author Contributions

**Conceptualization:** Takumi Kishimoto, Kenzo Okamoto, Kammei Rai, Yoichiro Kobashi.

**Data curation:** Takumi Kishimoto, Shigeki Koda, Mariko Ono, Yumi Umeda, Shotaro Yamano, Tomoki Takeda, Katsuya Kato, Yasumitsu Nishimura, Tetsuji Kawamura.

**Formal analysis:** Takumi Kishimoto, Kenzo Okamoto, Kammei Rai, Yoichiro Kobashi, Tetsuji Kawamura.

**Investigation:** Takumi Kishimoto.

**Methodology:** Takumi Kishimoto.

**Writing – original draft:** Takumi Kishimoto.

**Writing – review & editing:** Takumi Kishimoto.

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
