## [Decision Letter · Decision Letter 0]

1 Dec 2022

PONE-D-22-06467Respiratory disease in workers handling cross-linked water-soluble acrylic acid polymerPLOS ONE

Dear Dr. Kishimoto,

Thank you for submitting your manuscript to PLOS ONE. After careful consideration, we feel that it has merit but does not fully meet PLOS ONE’s publication criteria as it currently stands. Therefore, we invite you to submit a revised version of the manuscript that addresses the points raised during the review process. Please submit your revised manuscript by Jan 15 2023 11:59PM. If you will need more time than this to complete your revisions, please reply to this message or contact the journal office at plosone@plos.org. Please include the following items when submitting your revised manuscript:A rebuttal letter that responds to each point raised by the academic editor and reviewer(s). You should upload this letter as a separate file labeled 'Response to Reviewers'.A marked-up copy of your manuscript that highlights changes made to the original version. You should upload this as a separate file labeled 'Revised Manuscript with Track Changes'.An unmarked version of your revised paper without tracked changes. You should upload this as a separate file labeled 'Manuscript'.

We look forward to receiving your revised manuscript.

Kind regards,

Shama Ahmad, Ph.D.

Academic Editor

PLOS ONE

Journal Requirements:

^2. We note that the grant information you provided in the ‘Funding Information’ and ‘Financial Disclosure’ sections do not match. ^

^When you resubmit, please ensure that you provide the correct grant numbers for the awards you received for your study in the ‘Funding Information’ section.^

^3. ^Your ethics statement should only appear in the Methods section of your manuscript. If your ethics statement is written in any section besides the Methods, please delete it from any other section.

Reviewers' comments:

Reviewer's Responses to Questions

**Comments to the Author**

1. Is the manuscript technically sound, and do the data support the conclusions?

Reviewer #1: Yes

Reviewer #2: Yes

2. Has the statistical analysis been performed appropriately and rigorously? 

Reviewer #1: Yes

Reviewer #2: I Don't Know

3. Have the authors made all data underlying the findings in their manuscript fully available?

Reviewer #1: Yes

Reviewer #2: Yes

4. Is the manuscript presented in an intelligible fashion and written in standard English?

Reviewer #1: Yes

Reviewer #2: Yes

5. Review Comments to the Author

Reviewer #1: Thank you for giving me this opportunity to review this article. This indeed is an interesting study and not much is known about lung disorders caused by inhalation to acrylic polymers. Perhaps , we require more of animal studies and awareness regarding this particular concept. Few comments, though.

1)Were any blood investigations like ANA, RA factor, ACE levels done in these patients to exclude other kinds of ILDs?

2)Were these patients on any drugs/ receive any radiation in the past that could have lead to an ILD? Were these included in the questionnaire?

3) was a pre employment x ray , PFT done in these patients?

4) Were these patients subjected to PFT to know whether they have a restriction, obstruction or mixed pattern?

5) Did any of these patients develop any skin allergies in addition to the respiratory problems?

6)What were the shift hours of the patient like?

What kind of protective masks were they using?

What was the kind of ventilation in the rooms that they were working in?What the smoking index in the patients and was there any correlation with the progression?

Do you think these could have also made an impact on the severity of their problems?

7) Were these patients better on their off days i. e. better on their non working days?

Regards,

Dr Supriya.

Reviewer #2: Abstract

Line 32- add space after above

Line 37-change this disease’s to such disease

Line 38-change inhaling organic substances to cross linked water-soluble acrylic acid polymer (other organic substances cause fibrosis, see line 52)

Introduction

Line 48-add ‘incidences of’ after “after”

Figure and Figure Legends

1. Legend to fig 1 seems incomplete. You may want to explain the tasks here

2. Is fig 2 a represenatative of all cases or just case 1 and you need to show more. Please highlight or point to the findings with arrows

3. Fig 3 needs to be labeled and all the findings indicated in the legend needs to be indicated on the figures. Give subheading/ labels to each picture in the figure like in figure 4

4. Figure legend should have same labels like 4A not 4a. Indicate the findings with arrows

5. Capitalize the subheading in the legend. Red circular arrow is blurred and demonstrate other findings with arrows

6. Are both 6A and 6B interlobular septal thickening as mentioned in the legend. What do red circle indicate

7. Figure 8 is unclear what are the two scans. What is the box for. What do the arrows indicate

8. Carefully check each figure and legend to improve clarity.

6. PLOS authors have the option to publish the peer review history of their article (what does this mean?). If published, this will include your full peer review and any attached files.

Reviewer #1: No

Reviewer #2: No

---

## [Author Response · Author response to Decision Letter 0]

5 Jan 2023

December 22, 2022

Emily Chenette

Editor-in-Chief

Plos One

Dear Editor:

Thank you for allowing us to submit a revised draft of the manuscript entitled “Respiratory disease in workers handling cross-linked, water-soluble acrylic acid polymer” for publication in Plos One. We appreciate the time and effort that you and the reviewers dedicated to providing feedback on our manuscript and are grateful for the insightful comments on and valuable improvements to our paper. We have uploaded a copy of the original manuscript marked with all the changes made during the revision process. The revisions are shown as track changes in the manuscript file. Also appended to this letter are point-by-point responses to the comments raised by the reviewers. 

Response: Thank you for your remark. We have checked and confirmed that our manuscript meets these requirements.

Response: Per the editor's suggestion, we have corrected the appropriate sections. This study was not funded by a commercial company and was financially supported by a grant-in-aid from the Japan Organization of Occupational Health and Safety (Collaborative Research). There is no grant number in this research fund.

Response: We have made the appropriate changes.

Reviewers' Comments to the Authors:

5. Review Comments to the Author

Reviewer 1

1) Were any blood investigations like ANA, RA factor, ACE levels done in these patients to exclude other kinds of ILDs?

Author response: Thank you for your questions. All of ANA, RA factor, ACE are negative.

2) Were these patients on any drugs/ receive any radiation in the past that could have lead to an ILD? Were these included in the questionnaire?

Author response: There were no drugs and radiation for all cases.

3) was a pre employment x ray, PFT done in these patients?

Author response: The pre-employment X-ray was normal and no PET was done.

4) Were these patients subjected to PFT to know whether they have a restriction, obstruction or mixed pattern?

Author response: The restriction pattern is 5 and mixed pattern 1, normal 2.

5) Did any of these patients develop any skin allergies in addition to the respiratory problems?

Author response: The patients have no skin allergies.

6) What were the shift hours of the patient like?

Author response: They work for eight hours.

What kind of protective masks were they using?

Author response: Anti-dust exchangeable masks were used.

What was the kind of ventilation in the rooms that they were working in?

Author response: A device for exhausting dust was present in the rooms.

What the smoking index in the patients and was there any correlation with the progression?

Author response: The smoking index was 200 for Case 1, 460 for Case 2, 260 for Case 3, and 500 for Case 4.

Do you think these could have also made an impact on the severity of their problems?

Author response: Very high concentrations of dust may induce these problems of pulmonary and pleural diseases.

7) Were these patients better on their off days i. e. better on their non-working days?

Author response: These patients do not get better on their off days.

Reviewer 2

Abstract

Line 32- add space after above

Author response: Thank you for your comment. We have made the required change.

Line 37-change this disease’s to such disease

Author response: We have made the required change.

Line 38-change inhaling organic substances to cross linked water-soluble acrylic acid polymer (other organic substances cause fibrosis, see line 52)

Author response: We have made the appropriate changes.

Introduction

Line 48-add ‘incidences of’ after “after”

Author response: We have made the appropriate changes.

Figure and Figure Legends

1. Legend to fig 1 seems incomplete. You may want to explain the tasks here

Author response: We have added the required description.

2. Is fig 2 a represenatative of all cases or just case 1 and you need to show more. Please highlight or point to the findings with arrows

Author response: Thank you for your insightful comment. A case was added to Figure 4.

3. Fig 3 needs to be labeled and all the findings indicated in the legend needs to be indicated on the figures. Give subheading/ labels to each picture in the figure like in figure 4

Author response: Arrows or round arrows have been added to the figure.

4. Figure legend should have same labels like 4A not 4a. Indicate the findings with arrows

Author response: Our findings are indicated by arrows.

5. Capitalize the subheading in the legend. Red circular arrow is blurred and demonstrate other findings with arrows

Author response: We have made the appropriate changes.

6. Are both 6A and 6B interlobular septal thickening as mentioned in the legend. What do red circle indicate

Author response: We have made the appropriate changes for clarity.

7. Figure 8 is unclear what are the two scans. What is the box for. What do the arrows indicate

Author response: We have added the missing information.

8. Carefully check each figure and legend to improve clarity.

Author response: We have rechecked figures and legends and made the required changes.

We believe that the findings of this study are relevant to the scope of your journal and will be of interest to its readership. The manuscript has been carefully reviewed by an experienced editor whose first language is English and who specializes in editing papers written by scientists whose native language is not English.

We look forward to hearing from you at your earliest convenience.

Sincerely,

Takumi Kishimoto

Director of Research and Training Center for Asbestos-Related Diseases

1-10-25 Chikko Midorimachi, Minami-ku, Okayama 702-8055 Japan

Phone No: +81-86-262-9166

Fax No: +81-86-280-2828

Email Address: nakisimt@okayamah.johas.go.jp

---

## [Decision Letter · Decision Letter 1]

18 Jan 2023

PONE-D-22-06467R1Respiratory disease in workers handling cross-linked water-soluble acrylic acid polymerPLOS ONE

Dear Dr. Kishimoto,

Thank you for submitting your manuscript to PLOS ONE. After careful consideration, we feel that it has merit but does not fully meet PLOS ONE’s publication criteria as it currently stands. Therefore, we invite you to submit a revised version of the manuscript that addresses the points raised during the review process. The response to the review needs to be incorporated in the manuscript and provide where in the manuscript you have included the response. Remove the response letter from supplementary material and provide it where indicated during submission.

We look forward to receiving your revised manuscript.

Kind regards,

Shama Ahmad, Ph.D.

Academic Editor

PLOS ONE

Journal Requirements:

Reviewers' comments:

Reviewer's Responses to Questions

**Comments to the Author**

1. If the authors have adequately addressed your comments raised in a previous round of review and you feel that this manuscript is now acceptable for publication, you may indicate that here to bypass the “Comments to the Author” section, enter your conflict of interest statement in the “Confidential to Editor” section, and submit your "Accept" recommendation.

Reviewer #1: All comments have been addressed

Reviewer #2: All comments have been addressed

2. Is the manuscript technically sound, and do the data support the conclusions?

Reviewer #1: Yes

Reviewer #2: Yes

3. Has the statistical analysis been performed appropriately and rigorously? 

Reviewer #1: Yes

Reviewer #2: I Don't Know

4. Have the authors made all data underlying the findings in their manuscript fully available?

Reviewer #1: Yes

Reviewer #2: Yes

5. Is the manuscript presented in an intelligible fashion and written in standard English?

Reviewer #1: Yes

Reviewer #2: Yes

6. Review Comments to the Author

Reviewer #1: all comments have been addressed and the manuscript may be accepted for publication if there are no other issues.

Reviewer #2: Its unclear if my comments and their explanations are incorporated in the manuscript. The responses need to be specified where in the manuscript they have shown the information.

Also the response to comments is included as a supplementary file. It should be removed from there and included as a separate response letter.

7. PLOS authors have the option to publish the peer review history of their article (what does this mean?). If published, this will include your full peer review and any attached files.

Reviewer #1: No

Reviewer #2: No

---

## [Author Response · Author response to Decision Letter 1]

27 Mar 2023

February 10, 2023

Emily Chenette

Editor-in-Chief

Plos One

Dear Editor:

Thank you for allowing us to submit a revised draft of the manuscript entitled “Respiratory disease in workers handling cross-linked water-soluble acrylic acid polymer” for publication in Plos One. We appreciate the time and effort that you and the reviewers dedicated to providing feedback on our manuscript and are grateful for the insightful comments on and valuable improvements to our paper. We have uploaded a copy of the original manuscript marked with all the changes made during the revision process. The revisions are shown as track changes in the manuscript file. Also appended to this letter are point-by-point responses to the comments raised by the reviewers. 

Response: Thank you for your remark. We have checked and confirmed that our manuscript meets these requirements.

Response: Per the editor's suggestion, we have corrected the appropriate sections. This study was not funded by a commercial company and was financially supported by a grant-in-aid from the Japan Organization of Occupational Health and Safety (Collaborative Research). There is no grant number in this research fund.

Response: We have made the appropriate changes.

Reviewers' Comments to the Authors:

5. Review Comments to the Author

Reviewer 1

1) Were any blood investigations like ANA, RA factor, ACE levels done in these patients to exclude other kinds of ILDs?

Author response: Thank you for your questions. All of ANA, RA factor, ACE are negative.

2) Were these patients on any drugs/ receive any radiation in the past that could have lead to an ILD? Were these included in the questionnaire?

Author response: There were no drugs and radiation for all cases.

3) was a pre employment x ray, PFT done in these patients?

Author response: The pre-employment X-ray was normal and no PET was done.

4) Were these patients subjected to PFT to know whether they have a restriction, obstruction or mixed pattern?

Author response: The restriction pattern is 5 and mixed pattern 1, normal 2.

5) Did any of these patients develop any skin allergies in addition to the respiratory problems?

Author response: The patients have no skin allergies.

6) What were the shift hours of the patient like?

Author response: They work for eight hours.

What kind of protective masks were they using?

Author response: Anti-dust exchangeable masks were used.

What was the kind of ventilation in the rooms that they were working in?

Author response: A device for exhausting dust was present in the rooms.

What the smoking index in the patients and was there any correlation with the progression?

Author response: The smoking index was 200 for Case 1, 460 for Case 2, 260 for Case 3, and 500 for Case 4.

Do you think these could have also made an impact on the severity of their problems?

Author response: Very high concentrations of dust may induce these problems of pulmonary and pleural diseases.

7) Were these patients better on their off days i. e. better on their non-working days?

Author response: These patients do not get better on their off days.

Reviewer 2

Abstract

Line 32- add space after above

Author response: Thank you for your comment. We have made the required change.

Line 37-change this disease’s to such disease

Author response: We have made the required change.

Line 38-change inhaling organic substances to cross-linked water-soluble acrylic acid polymer (other organic substances cause fibrosis, see line 52)

Author response: We have made the appropriate changes (New line 39).

Introduction

Line 48-add ‘incidences of’ after “after”

Author response: We have made the appropriate changes (New line 50).

Figure and Figure Legends

1. Legend to fig 1 seems incomplete. You may want to explain the tasks here

Author response: The details of the work was described in detail (please refer to the legend of figure 1).

2. Is fig 2 a representative of all cases or just case 1 and you need to show more. Please highlight or point to the findings with arrows

Author response: Thank you for your insightful comment. We added case 2, a typical case similar to case 1, as figure 4. In figures 2 and 4, we inserted arrows and round arrows to indicate points. For figure 4, we added an explanatory note.

3. Fig 3 needs to be labeled and all the findings indicated in the legend needs to be indicated on the figures. Give subheading/ labels to each picture in the figure like in figure 4

Author response: Arrows or round arrows have been added to the figure. We labeled figures 3A, 3B, 3C, and 3D and inserted black arrows (large and small) and white arrows to indicate the findings. We have also added an explanatory note.

4. Figure legend should have same labels like 4A not 4a. Indicate the findings with arrows

Author response: We have newly labeled the figure as Fig. 5A and 5B, inserted white arrows to indicate the findings, and each explanatory note is the original.

5. Capitalize the subheading in the legend. Red circular arrow is blurred and demonstrate other findings with arrows

Author response: We deleted the red circular arrow and indicated findings using arrows and round arrows. The Explanatory note is the original.

6. Are both 6A and 6B interlobular septal thickening as mentioned in the legend. What do red circle indicate

Author response: We deleted the red circular arrow in figures 7A and 7B and indicated findings using arrows and round arrows. The Explanatory note is the original.

7. Figure 8 is unclear what are the two scans. What is the box for. What do the arrows indicate

Author response: The two scans are shown in new figures 9 and 11, with one scan per figure. We deleted the arrows and the explanatory note for the lesioned area is the original.

8. Carefully check each figure and legend to improve clarity.

Author response: We have rechecked figures and legends and made the required changes.

9. New figures 10, 12, and 13 are presented in order of magnification, and to make it easier to view the EW staining, it was presented with decreasing magnification.

We believe that the findings of this study are relevant to the scope of your journal and will be of interest to its readership. The manuscript has been carefully reviewed by an experienced editor whose first language is English and who specializes in editing papers written by scientists whose native language is not English.

We look forward to hearing from you at your earliest convenience.

Sincerely,

Takumi Kishimoto

Director of Research and Training Center for Asbestos-Related Diseases

1-10-25 Chikko Midorimachi, Minami-ku, Okayama 702-8055 Japan

Phone No: +81-86-262-9166

Fax No: +81-86-280-2828

Email Address: nakisimt@okayamah.johas.go.jp

---

## [Editor Report · Decision Letter 2]

11 Apr 2023

Respiratory disease in workers handling cross-linked water-soluble acrylic acid polymer

PONE-D-22-06467R2

Dear Dr. Kishimoto,

We’re pleased to inform you that your manuscript has been judged scientifically suitable for publication and will be formally accepted for publication once it meets all outstanding technical requirements.

Kind regards,

Shama Ahmad, Ph.D.

Academic Editor

PLOS ONE
---

## [Editor Report · Acceptance letter]

13 Apr 2023

PONE-D-22-06467R2 

Respiratory disease in workers handling cross-linked water-soluble acrylic acid polymer 

Dear Dr. Kishimoto:

I'm pleased to inform you that your manuscript has been deemed suitable for publication in PLOS ONE. Congratulations! Your manuscript is now with our production department. 

Kind regards, 

on behalf of

Dr. Shama Ahmad 

Academic Editor

PLOS ONE